# Mapping career patterns in research: A sequence analysis of career histories of ERC applicants

**Claartje J. Vinkenburg**[1]*, **Sara Connolly**[2], **Stefan Fuchs**[3], **Channah Herschberg**[4], **Brigitte Schels**[3,5]

**1** Independent Expert, Affiliated with Vrije Universiteit Amsterdam, Amsterdam, The Netherlands, **2** Norwich Business School, University of East Anglia, Norwich, United Kingdom, **3** Institute for Employment Research, Nuremberg, Germany, **4** Institute for Management Research, Radboud University, Nijmegen, The Netherlands, **5** Friedrich-Alexander University of Erlangen-Nuremberg, Nuremberg, Germany

☉ These authors contributed equally to this work.
* c.j.vinkenburg@gmail.com

**Data Availability Statement:** Unrestricted and uncontrolled access to the complete career history data (in terms of position, institution, contract type, location etc. of all spells since PhD) compromises the confidentiality and privacy of research

## Abstract

Despite the need to map research careers, the empirical evidence on career patterns of researchers is limited. We also do not know whether career patterns of researchers can be considered conventional in terms of steady progress or international mobility, nor do we know if career patterns differ between men and women in research as is commonly assumed. We use sequence analysis to identify career patterns of researchers across positions and institutions, based on full career histories of applicants to the European Research Council frontier research grant schemes. We distinguish five career patterns for early and established men and women researchers. With multinomial logit analyses, we estimate the relative likelihood of researchers with certain characteristics in each pattern. We find grantees among all patterns, and limited evidence of gender differences. Our findings on career patterns in research inform further studies and policy making on career development, research funding, and gender equality.

*Data on the career paths of young researchers would help [. . .]. There is a pressing need for greater transparency about the likelihood of PhD students and postdocs following an academic career to the higher levels. [. . .]. vn[1].*

## Introduction

The need to map research careers is tied to policy efforts to stimulate career mobility and enhance career development for researchers [2–4], with the ultimate goal to strengthen innovation and the knowledge economy. However, despite some efforts to map research careers in the European context [5–9], exactly *how* research careers develop in terms of patterns or

participants, and violates the conditions on ethics approval obtained for this study from the Ethical Committee of the European Research Council. Simplified de-identified data sets that contain minimal but relevant personal (age, gender, children, etc) and career related variables, including a career pattern denominator, are available upon request. The data sets are available through the University of East Anglia: https://people.uea.ac.uk/en/datasets/mapping-career-patterns-in-research-a-sequence-analysis-of-career-histories-of-erc-applicants(a64c76cc-da8f-4ab1-b19f-7a3b3a814d7f).html. Please contact openresearch@uea.ac.uk to explain why you need the data and purposes for which they will be used. The data will be made available through one of the beneficiaries of our ERCAREER grant, Professor Sara Connolly.

**Funding:** This work was supported by the European Research Council (ERC https://erc.europa.eu/) Coordination and Support Action (CSA) [ERC-CSA-2012-317442], project acronym ERCAREER, awarded to CJV SC SF. The funder was instrumental in data collection.

**Competing interests:** The authors have declared that no competing interests exist

moves through positions and institutions remains largely uncharted territory [1]. Research careers are often described in terms of outcomes (i.e. publications) [10] or mobility events (i.e. international moves) [11]. Following Abbott, we view the career pattern itself as an outcome [12]. After obtaining a PhD, researchers move through job positions within and between institutions. From a holistic life course perspective, careers are not (only) marked by singular specific events but also a sequence of states that may differ in progression and timing [13]. However, details on differences in career trajectories of individual researchers are lacking [14]. Based on full career histories of European Research Council (ERC) Starting and Advanced grant applicants, we contribute to earlier studies of research careers by mapping the career patterns of men and women researchers from their PhD to more established careers. We do not start from theoretical or anecdotal assumptions about career patterns but use a relatively new analytical strategy developed to empirically capture the nature of career patterns over time and place, providing an overview of research careers in different disciplinary and national settings across Europe.

Career patterns can be interpreted as objectively observable paths of movement through occupational hierarchies [15]. However, despite the ubiquitous presence of the term "career patterns" in discourse and writings about careers in research, earlier efforts to track research careers yielded limited evidence on exactly how research careers develop over time. We often assume that researchers follow a very similar and traditional career path after obtaining their PhD degree [16]. The normative expectation of upward mobility has changed from a stylized career path [17] based on a very limited number of academic "rites of passage" (e.g. PhD defense, inaugural lecture) toward a new career model of cumulative promotions [18]. However, such expectations and assertions are rarely built on an evidence base of actual career patterns in research. Our analysis reveals how research careers develop over time, in terms of moving through positions and institutions, and whether career patterns beyond the "traditional" can be identified among researchers who apply to the ERC.

The ERC in looking for "excellence only" aims at selecting "groundbreaking" and "truly novel research" for funding [19,20]. By funding and thus organizing excellent science at the European level [21–23], the ERC extends national funding schemes with unique conditions: generous, long-term, flexible, and risk-tolerant [24]. The ERC's prestigious individual research grants [20,25] are awarded based on a peer-reviewed evaluation of the quality of the principal investigator and the research proposal [19]. Similar to other grant schemes, ERC evaluators rank applications taking into account both the science and the scientist [26,27]. The career histories of applicants, thus, play an important role in the ERC peer review process. Previous studies have shown that funded applicants (grantees) and non-funded applicants in various research funding schemes do not differ (much) on objective quality criteria [28,29] and therefore we include both funded and non-funded applicants in our study. Applications to the ERC are made through a host institution, where the research will be undertaken, and there is typically an internal sorting within institutions resulting in support for only the highest quality applications [30,31]. We therefore argue that both the funded and non-funded applicants are among the most excellent researchers of their generation as their applications have been submitted to the most prestigious European research funding organization. Using an exploratory, empirical approach we study how the careers of these researchers develop and whether they develop in a similar manner–in accordance with the assumed traditional career path in research and matching normative expectations of upward mobility.

In addition, we study another commonly held assumption, namely that the careers of men and women in research tend to develop differently. In their initial report on research careers in Europe, ESF [2] states that "almost all obstacles and bottlenecks identified during a research career affect the careers of women scientists more severely than those of men", with the main

underlying cause of this difference being care responsibilities, which fall disproportionally to women. This assumption is found extensively in the literature and also resonates in the call for proposals sent out by the ERC gender balance thematic working group in 2011 to map "the paths and patterns, differences and similarities in the career paths of women and men ERC grantees". Our proposal was selected by the ERC to explore gender aspects in career structures and career paths of applicants.

However, despite women's relative underrepresentation at the highest levels in most research fields [32], and given that women ERC grantees have lower publication rates than men [33], we do not know whether women researchers' career develop at a different speed or in a different way than men's, nor do we know the actual impact of care responsibilities on career patterns. We therefore empirically test the likelihood of men and women following different career patterns, as well as the extent to which certain personal and institutional characteristics affected this likelihood differentially for men and women.

To map career patterns in research across disciplines around Europe, we use a specific kind of sequence analysis called Optimal Matching Analysis (OMA). OMA incorporates timing alongside transition between occupational states, offering an appropriate analytical tool for the study of careers [34,35]. Abbott [36,37] proposed using OMA, as an appropriate method for measuring life courses "as they are", calling this descriptive approach a paradigm shift from causes to events. OMA is used to identify order in sequences by analyzing the similarity of sequences to one another and sorting them into groups of similar sequences [13]. Using data on career histories of ERC grant applicants, OMA provides insights into career patterns among early and established researchers, highlighting differences and similarities. For each grant scheme we identify patterns reflecting combinations of positional and institutional sequences, different progression logics, and movements–including leave or spells of unemployment. In distinguishing five career patterns for early and five for established researchers across Europe, we explore whether certain patterns are more common or "conventional" than others, whether some patterns are associated with greater likelihood of application success, and how gender and other personal, disciplinary, and PhD-related factors affect the likelihood and appearance of career patterns. This mapping of research career patterns should inform research policy, in terms of promoting career development, mobility, and gender equality in funding.

## Career patterns in research

The origins of the construct of career patterns can be found in industrial sociology where "it was viewed, objectively, as the number, duration, and sequence of jobs in the work history of individuals" [38,39]. Career conventions, or general agreements on descriptions of common career patterns, are likely to be normative, in the sense that they provide prescriptions of what careers in research *should* look like. The notion of an ideal career in research likely translates into career conventions in terms of linearity or steady progress [40], early successes [41], institutional prestige [42], and (inter) national mobility [43,44].

These conventions have been surprisingly stable despite the increasing demographic diversity of those who do research and the challenges to the conventional view of research careers associated with this diversity, most notably perhaps with respect to the representation of women [32,45,46]. It is evident that career conventions matter in selection decisions (including funding). Decision makers use signals such as linearity and mobility (upward and across borders), sometimes even as a proxy for excellence [44,47]. Careers as represented by CVs play an important part when funding decisions are made [26,27,48,49] and are viewed through lenses that are affected by the context, culture, and gender of the candidate and the evaluator.

Knowledge of career progress in terms of moving between positions within and across different types of institutions (e.g., universities, research institutes) is important for the evaluation of researchers' standing and independence [25].

Despite more than a decade of efforts to track research careers across disciplinary and national contexts, conclusive answers on career patterns of researchers are missing. To gain insight into the existing empirical evidence on career patterns in research, we performed an extensive literature review (see S1 File for search strategy and detailed findings; and [14] for an earlier version of the review). From the final set of 40 peer reviewed sources, we conclude that the number of existing empirical studies that shed light on what career patterns in research "objectively" look like is very small. While many sources refer to the existence of "career patterns", there are actually only *three* studies that empirically distinguish unique patterns in research based on temporal combinations of positions and institutions. Two of these use CVs to identify distinct career patterns for senior administrators in U.S. universities [34,50]. The third is a recent paper [51], which differentiates five early career patterns based on narratives from young academics crossing disciplinary, institutional, and national borders.

The majority of the sources reviewed in fact do *not* distinguish patterns, but rather characteristics of careers, predictors of career advancement, or mobility events. What authors call "patterns" are typically counts of mobility events collected from CVs or surveys. Most sources by virtue of their data are limited to one location or one discipline. The dominant theme is gaining an understanding of how (international) mobility, early success (e.g. grants), publications and/or citations contribute to promotion, prestige, and income. A second dominant theme is gender [52,53]. The common assumption that women's careers in research are less likely than men's to resemble an uninterrupted linear pattern, due to women's typically larger share in care responsibilities, is both a rationale for *and* a finding of studies looking at gender in research career. Gender differences in career advancement or representation are hypothesized to result from gender gaps in publication or mobility. Given that career indicators are used to evaluate grant applications, the finding that women receive lower evaluations on their "quality of researcher" assessment than men [26,27] may reflect both a greater actual diversity in career patterns amongst women than men *and* assumptions made about such diversity in career patterns.

In conclusion, our literature review (see S1 File) reveals a profound disconnect between compelling notions of what a conventional career in research looks like, and the lack of insight into the appearance and frequency of "actual" career patterns in research across distinct institutional, disciplinary, and national contexts. Our study sheds light on the reality behind normative career expectations and conventions, and takes a holistic view across different contexts. Based on the limited empirical evidence on research careers, we test the likelihood of following a particular career pattern depending on the context (in- or outside academia, institutional prestige) and personal characteristics (gender, children, cohort).

## Data and methods

### Research context

The European Research Council (ERC) established its grant schemes in 2007 in order to "support investigator-driven frontier research across all fields, on the basis of scientific excellence" [54]. The Starting Grant scheme (StG) was intended for researchers up to 12 years after their PhD, with subcategories for "starters" (within 7 years of the PhD) and "consolidators" (8–12 years after the PhD). Since 2013, the Starting Grant scheme has been divided into the separate Starting and Consolidator grants, but at the time we collected our data, this was a single scheme. The Advanced Grant scheme (AdG) is aimed at established researchers with a strong

research record who are considered to be leaders in their field. Funding entails a long-term, individual grant in order to conduct groundbreaking, curiosity-driven, high-risk high-gain research–from 1.5 to 2.5 million Euros. Applications are accepted across disciplines and reviewed by expert sub-panels within the umbrella of three domains: LS–life sciences; PE–physical sciences and engineering; SH–social sciences and humanities (details in S3 File).

## Participants and procedure

We used data on individual career histories that we collected in a survey of ERC applicants. The advantage of the survey is that respondents were directly asked whether they experienced career interruptions such as unemployment, parental leave. These career breaks may be under-reported in their CVs. Due to data protection regulations, the ERC gave us permission to survey those who had applied for the StG in 2012 –as applicants from previous years were surveyed as part of an earlier ERC funded project [55]–and all AdG applicants between 2007 and 2012. Therefore, our potential sample comprised applicants who gave consent for the use of their data at the time of application to the ERC (33% of StG and 39% of AdG applicants). Our data collection and protection procedures were described in the declaration on ethics considerations of ERC-CSA-2012-317442 ERCAREER, approved by the ERC Executive Agency, in compliance with the terms of Regulation EC 45/2001, and included written consent of survey participants.

The surveys were constructed to collect data on the paths that researchers take from PhD to their current position. The survey design for StG and AdG applicants was slightly different, to reflect the relative length and complexity of the career. For both surveys, we included questions on job positions and institutional affiliations of all spells of employment after completing the PhD, as well as other states, such as unemployment and different types of leave. To account for differences in career length and complexity, the survey for the AdG applicants started with the first job position, for StG applicants directly after the PhD. The surveys also included questions on reasons for mobility or changes in position, family situation, parental leave and other career breaks, perceived institutional support, and career aspirations (StG only). For replication purposes, a pdf version of the online surveys is provided (see S2 File). The information collected via the surveys was matched with information provided by the applicant on their application form (contact information, host institution, gender, nationality, year of PhD) and some administrative information (sub-domain, application outcome) provided by the ERC.

A personalized email invitation with a link to an online survey was sent in October and November 2013 via email to 1,588 StG 2012 applicants (460 women, 29%) and to 4,088 AdG applicants (632 women, 15.46%) from the cohorts 2008 to 2012. Respondents who did not finish the survey were excluded from the analysis. For our analysis of career patterns, we used 322 completed responses from StG applicants (20% response rate, 126 from women, 39%) and 737 completed responses from AdG applicants (18% response rate, 145 from women, 20%). The StG 2012 and AdG applicant samples are representative of their respective populations in terms of discipline composition (see S1 File). However, funded grantees and women are over-represented in the sample, possibly because grantees may have felt an obligation to the ERC, and women may have been more motivated by the topic of the survey and thus more likely to respond to the invitation. We calculated probability weights relating the sample population with the ERC applicants' population based on gender, discipline and grantees, which we apply in our bivariate descriptive analysis. Weighting changes the share of women and grantees in each cluster; however, the findings are robust when comparing the results from unweighted or weighted data.

## Identifying research career patterns

In the first step of our analysis, we used Optimal Matching Analysis (OMA) with cluster analysis in order to identify and compare groups of typical research careers. Following this approach, we conceptualize the unfolding of careers as outcomes [56] reflecting researchers' trajectories through different positions and institutions. OMA is an exploratory method to identify patterns, in terms of sequences of states (position and succession) in longitudinal data [37], and, thus, a recommended analytical method in careers research [57] (see [13] for a critical overview of applications of OMA).

To model research careers, we defined ten positional and seven institutional states that capture the variance we are interested in. We include five different job positions that reflect differences in status, hierarchy and tasks: (1) Postdoc; (2) Lecturer; (3) Senior Lecturer; (4) Professor; and (5) Other job. Each of the categories also includes comparable job descriptions from different national and discipline-specific contexts. The categorization was based on a coding scheme that we developed from a preliminary analysis of 180 CVs of ERC applicants. It was cross-checked with existing European frameworks for research careers [4,58], (details in S1 File). While the position labels used in the analysis reflect common denominations in university settings (e.g. senior lecturer), the survey provided examples of equivalent labels used in non-university settings (e.g. senior researcher). The five other positional states were: (6) Unemployed; (7) Research leave; (8) Parental leave; (9) Other status (e.g., illness or military service); and (10) Gap, if no information is provided. We defined the following seven institutional states: (1) Universities and other institutions of higher education; (2) Non-profit research institutions; (3) Commercial research institutes; (4) Hospitals or clinics; (5) Government; (6) Private organizations; and (7) Other.

In the analysis, we identified the positional state and the institutional state for each person in each month from PhD to application for ERC grant. An example for three researchers A, B and C is given in Table 1. Each combination of numbers (e.g. 8–7) reflects a combination of position and institution. A and B have been in a postdoc position at a university (1–1) in the first month after PhD; after 36 months, A is a lecturer at a university (2–1) while B is on parental leave (8–7); after 72 months, A is a senior lecturer at another university (3–1) while B is a government policy officer (5–5). In contrast, C started in a job in a commercial research institute (5–2) after PhD and stayed in this job for several years, before they are, 72 months after PhD, in an executive position in the same institute (3–2). While the careers of A and B start in the same way, they develop differently. In contrast, the career of C runs through different positions from the beginning. OMA is an explorative method that allows to investigate whether there are comparable structures and differences within individual research careers that are aggregated to typical patterns.

**Table 1. Research career sequences, defined by positional state (first digit) and institutional state (second digit), for selected months since PhD–three illustrations.**

| Month since PhD | 1 | . . . | 36 | . . . | 72 | . . . |
|---|---|---|---|---|---|---|
| A | 1–1 | | 2–1 | | 3–1 | |
| B | 1–1 | | 8–7 | | 5–5 | |
| C | 5–2 | | 5–2 | | 3–2 | |

Positional state: (1) Postdoc; (2) Lecturer; (3) Senior Lecturer; (4) Professor; (5) Other job; (6) Unemployed; (7) Research leave; (8) Parental leave; (9) Other status; (10) Gap

Institutional state: (1) Universities/institutions of higher education; (2) Non-profit research institutions; (3) Commercial research institutes; (4) Hospitals/clinics; (5) Government; (6) Private organizations; (7) Other.

OMA compares each sequence in a sample with any other sequence and calculates distances between the sequences. To do so, OMA calculates the costs of transferring one sequence into another by deleting, inserting, or substituting the states of a sequence. Costs are assigned to each of these transformations. The OMA calculates the distance between any two sequences as the minimum possible costs of transformation and generates a matrix of distances for all sequence combinations. In other words, the distance between the sequences of two individuals is lower when fewer steps are needed to make them equal (as can be illustrated using the example in Table 1). For this analysis, we used the information on the sequence of job positions and institutional affiliations for each month after finishing the PhD until the date of application for the ERC grant, so that the length of the sequences varies between individuals. Due to the structure of the survey, the period of observation for the AdG applicants started with the first job position, for StG applicants directly after the PhD. Robustness test show that restricting the StG observation to the first job position would not change our results. The appearance of the career patterns is not driven by differences in career length (details in S3 File).

For our analysis, the sequences of job positions and institutional affiliations for each person were treated as two channels: in a first step of the analysis, the costs are specified separately for each channel: second, the substitution costs for each time point are aggregated in order to calculate a combined substitution cost matrix [59]. The costs for insertion and deletion ("indel costs") were set at 1 and substitution costs were set at 2. In our setting, substitution operations are as expensive as one insertion and one deletion operation so that they can be interchangeable in their use [57]. The calculated distances measures were normalized to account for differences in sequence lengths. We applied the OMA for each sample, StG and AdG separately, using the statistical software R and "TraMineR".

In order to identify the main typical career patterns after OMA, hierarchical Ward cluster analysis was used to group sequences according to their similarity based on the matrix of distances generated. Sequences bundled within a particular group are close to one another and distant to other sequences. From the cluster dendrograms (see S3 File), the space of meaningful distinctions and then the possible number of groupings were derived [60]. Furthermore, the grouping of sequences was chosen that offered the best explanatory power for the overall research questions [61]. For each sample, StG and AdG applicants, we identified five distinct clusters representing unique career patterns, described in the results section.

## Analyzing characteristics of researchers in the career patterns

In the second step of the analysis, we estimated which characteristics influence the likelihood of researchers belonging to one pattern using multivariate multinomial logistic regressions, estimated separately for StG and AdG samples. Results of the multinomial logistic regressions are presented in the results section. The five distinct career patterns were used as the dependent variable. We applied a robustness test with a two-step selection model to take account of possible response bias to the survey; this did not change our conclusions. Further details on this test are available upon request from the authors.

We were interested whether the likelihood of following a specific career pattern is associated with the research discipline, PhD-related characteristics, and personal characteristics. Descriptive statistics for relevant characteristics can be found in the S3 File). We included the broad disciplinary areas of the research captured by the ERC categorization between Life Sciences (LS), Physical Sciences and Engineering (PE), and Social Sciences and Humanities (SH). Regarding PhD-related factors, the research prestige of the PhD granting institution for all respondents was measured by assigning the 2014 "Leiden score" (the proportion of the publications of each research institution or university belonging to the top 10% of their field [62]).

A dummy variable was used to control for the cases for which we have no information on the Leiden score of their PhD institution. To control for career-specific factors preceding the period of observation, we controlled for work experience before the PhD and age at the time of the PhD. We considered care responsibilities by parenthood status and age of the youngest child (under the age of three or older). We strictly used information before obtaining a PhD to ensure clear interpretation of which researchers enter which pattern and not to mix up conclusions with outcomes of career processes, for example, parenthood during the period of observation. Personal characteristics include gender and birth cohort. Nationality of the applicants is an additional control variable.

## Results

### Identified career patterns

Each of the five distinct Starting Grant (StG) and Advanced Grant (AdG) career patterns represents a unique and temporal combination of positions and institutions. The cluster figures provided in Figs 1 and 2 illustrate the order and timing of job positions and institutional affiliations. The upper graphs plot the individual sequences of positions and affiliations for each observation in the cluster and, thus, illustrate the career complexity among researchers. The bottom graphs plot the monthly breakdown of the different status in each cluster. These figures provide an aggregate picture of the share of researchers in each job position and institutional affiliations and the change of these shares over the career progress. Tables 2 and 3 provide additional information on cluster characteristics.

**Starting grant career patterns.** In the first StG pattern (Fig 1), the postdoc position is concentrated at the beginning and lectureships at the end. The careers are predominantly located in universities. These movements reflect a career model of upward mobility and we label this pattern as *steady progress at universities*. In contrast, the second StG pattern shows a

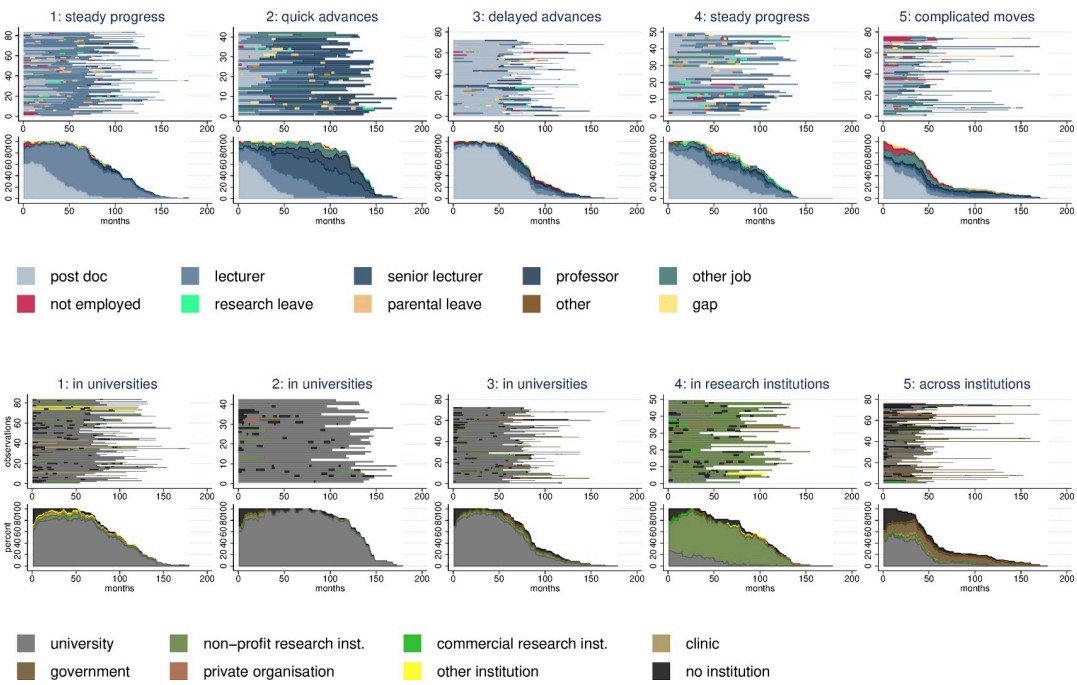

**Fig 1. Starting grant career patterns.** Sequence index plots and status proportion plots for positions and institutions.

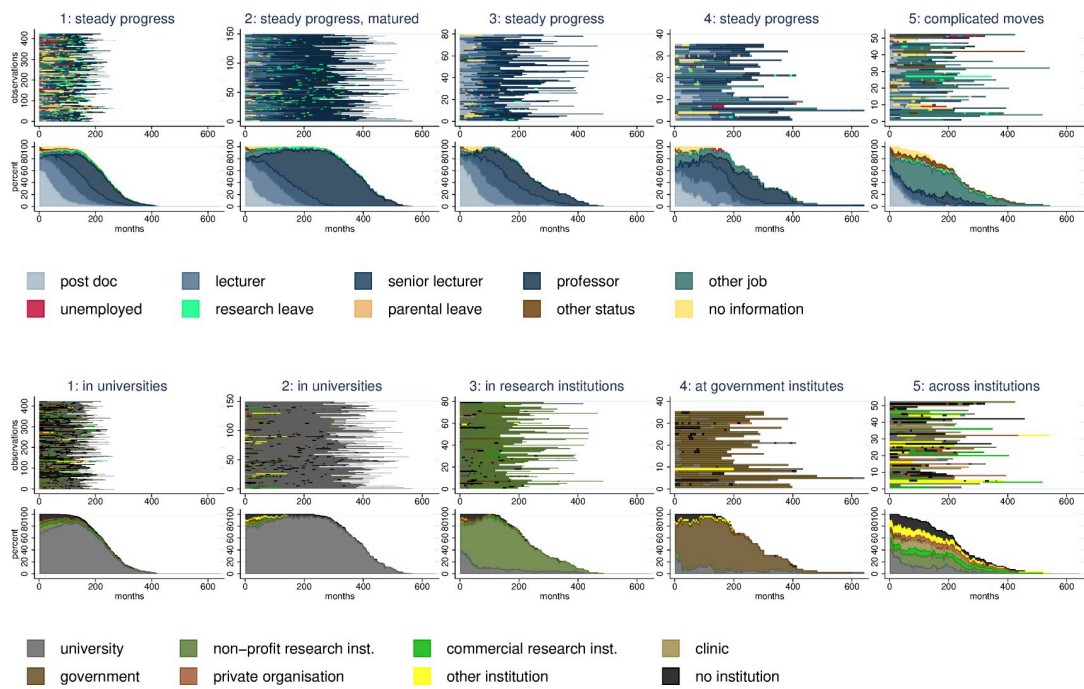

**Fig 2. Advanced grant career patterns.** Sequence index plots and status proportion plots for positions and institutions.

relative dense sequence of postdoc, lecturer, senior lecturer, and professor positions. Although this cluster has the longest average duration, the figures indicate that researchers move relatively rapidly from one position to the next up the hierarchy. Again, we find a relative stable institutional university affiliation–we label this pattern *quick advances in universities*. In the third StG pattern, the postdoc is the dominant position and this is relative stable over time. At the end of the observation period, relatively few researchers have moved to lecturer or senior lecturer positions. Again, the careers are predominantly located at universities and we label this pattern *delayed advances in universities*. In the StG fourth pattern, the postdoc position is concentrated at the beginning and lecturer at the end. The job position progression is very similar to those in our first cluster–steady progress in universities–but careers are predominantly located in research institutes. We identify this fourth distinct pattern as *steady progress in research institutions*. The final StG pattern consists of several job positions that are concentrated within relative short careers. Although postdoc positions are the most frequent status, the careers in this cluster also consist of higher shares of unemployment and other jobs compared to the other clusters. Researchers also moved across different institution types. Thus, the final pattern is labeled *complicated moves across institutions*.

Of the five StG career patterns, three account for more than 20% of the sample and 74% in total: *steady progress in universities* (27%), *complicated moves across institutions* (24%), and *delayed advances in universities* (23%).

Regarding the composition of the patterns (see descriptives in Table 2), we can see there are relatively more women in the *steady progress in research institutions* and *complicated moves across institutions* clusters than in the *quick advances in universities* pattern. The grant success rate varies between the different career patterns from 10 per cent in the *complicated moves across institution* pattern to 19 per cent in the *quick advances in universities* pattern. However, none of these differences are statistically significant. Excellence, it could be said, can be found in all career patterns.

**Table 2. Researchers in the Starting Grant (StG) patterns (descriptives, multinomial logistic regressions, average marginal effects).**

| | steady progress | | quick advances | | delayed advances | | steady progress | | complicated moves | | All | Test statistics |
|---|---|---|---|---|---|---|---|---|---|---|---|---|
| | *in universities* | | *in universities* | | *in universities* | | *in research institutes* | | *across institutions* | | | |
| n | 83 | | 42 | | 72 | | 49 | | 76 | | 322 | |
| ***Descriptives (weighted)*** | | | | | | | | | | | | | |
| % (row percentages) | 26.5 | | 13.6 | | 22.7 | | 13.6 | | 23.5 | | 100.0 | |
| mean length of sequences (months) | 100 | | 129 | | 89 | | 92 | | 62 | | 92 | 34.49*** [†] |
| *Characteristics (column percentages)* | | | | | | | | | | | | | |
| Women | 25.1 | | 19.3 | | 27.0 | | 38.5 | | 34.2 | | 28.7 | 5.71 [‡] |
| Successful grant applicants | 7.8 | | 18.7 | | 13.2 | | 17.1 | | 9.5 | | 12.1 | 4.87 [‡] |
| ***Estimates from Multinomial Logistic Regression*** | | | | | | | | | | | | | |
| | AME | p | AME | p | AME | p | AME | p | AME | p | | |
| *Discipline (Ref: LS)* | | | | | | | | | | | | | |
| PS | .046 | .390 | .028 | .459 | -.071 | .180 | -.080 | .106 | .075 | .154 | | |
| SH | .028 | .672 | .093 | .127 | -.023 | .736 | -.172 | .000 | .074 | .252 | | |
| *PhD related characteristics* | | | | | | | | | | | | | |
| Age when received PhD (centered) | .001 | .884 | -.020 | .015 | -.003 | .759 | .001 | .858 | .020 | .016 | | |
| PhD in home country | .087 | .134 | -.047 | .343 | -.010 | .874 | -.000 | .999 | -.029 | .646 | | |
| Leiden score of PhD Institution | -.003 | .780 | .005 | .478 | .011 | .230 | .007 | .333 | -.020 | .040 | | |
| *Age of youngest child when received PhD (Ref,: no children)* | | | | | | | | | | | | | |
| under 3 | -.095 | .243 | .131 | .114 | -.029 | .714 | .058 | .469 | -.065 | .428 | | |
| 3 or older | -.262 | .000 | .032 | .686 | .259 | .088 | .055 | .641 | -.084 | .321 | | |
| Work exp. Before PhD | -.037 | .537 | -.081 | .047 | .046 | .467 | .036 | .391 | .010 | .843 | | |
| *Personal characteristics* | | | | | | | | | | | | | |
| Female | -.002 | .964 | -.100 | .003 | -.025 | .593 | .078 | .074 | .050 | .311 | | |
| Born after 1970 | -.039 | .642 | -.444 | .000 | .223 | .000 | .110 | .008 | .150 | .003 | | |
| *Nationality (ref: Central Europe)* | | | | | | | | | | | | | |
| East Europe | -.182 * | .017 | .096 | .283 | -.095 | .249 | .009 | .910 | .173 | .127 | | |
| Southern Europe | -.056 | .376 | .039 | .386 | -.038 | .542 | .065 | .238 | -.011 | .852 | | |
| Northern Europe | -.083 | .405 | .141 | .092 | .004 | .972 | -.046 | .495 | -.016 | .868 | | |
| Rest world | .107 | .255 | .021 | .708 | -.067 | .371 | .-001 | .991 | -.060 | .421 | | |
| Pseudo R2 | .107 | | | | | | | | | | | |

Ref.: reference category, LS.: Life sciences, PS.: Physical sciences and Engineering, SH.: Social sciences and Humanities; control variables for no information for age at PhD and Leiden score of PhD institution; Anova test statistics [†], Chi2 test statistics [‡] Sig.:

\*\*\* p<0.001.

\*\* p<0.01.

\* p<0.05. + p< = 0.1

**Advanced grant career patterns.** In the first AdG career pattern (Fig 2), the postdoc position is concentrated at the beginning followed by lecturer, senior lecturer, and professor. The careers are predominantly located at universities; we label this pattern (as we do in for the StG) *steady progress at universities*. The second AdG career pattern is similar to the first, but reflects longer careers at universities and especially in professorial positions. We label this pattern *mature progress in universities*, as it reflects steady progress that has reached maturity or even a ceiling. The sequence of job positions in the third cluster is similar to the first cluster but is predominantly located in research institutes. We label this pattern *steady progress at research institutes*. The fourth AdG pattern reflects steady progress from one position to the next, including a higher share of

'other' job positions, within government institutions. We label this pattern *steady progress in government institutes*. Finally, in the fifth AdG pattern, we again find complicated moves across various institutional settings. Many individuals in this cluster start with postdoc positions, moves to other jobs, and this cluster consists of significantly more states, including unemployment, than others. Thus, the final pattern is labeled *complicated moves across institutions*.

In comparison to the composition of the StG sample, there is greater coherence amongst the AdG applicants, with over half belonging to a single pattern–*steady progress at universities* (57%). The two university based career patterns (*steady* and *mature progress*) account for over three-quarters of the entire sample (see descriptives in Table 3). Across different clusters, there

**Table 3. Researchers in the Advanced Grant (AdG) patterns (descriptives, multinomial logistic regressions, average marginal effects).**

| | steady progress | | steady progress, matured | | steady progress | | steady progress | | complicated moves | | All | Test statistics |
|---|---|---|---|---|---|---|---|---|---|---|---|---|
| | in universities | | in universities | | in research institutes | | at government institutes | | across institutions | | | |
| n | 422 | | 149 | | 79 | | 35 | | 52 | | 737 | |
| *Descriptives (weighted)* | | | | | | | | | | | | |
| % (row percentages) | 56.8 | | 20.4 | | 10.8 | | 4.9 | | 7.2 | | 100.0 | |
| mean length of sequences (months) | 242 | | 396 | | 264 | | 292 | | 240 | | 278 | 104.12*** † |
| *Characteristics (column percentages)* | | | | | | | | | | | | |
| Women | 15.8 | | 7.1 | | 18.5 | | 20.0 | | 16.4 | | 14.6 | 9.17* ‡ |
| Successful grant applicants | 13.8 | | 15.5 | | 13.5 | | 8.6 | | 9.3 | | 13.5 | 2.06 ‡ |
| *Estimates from Multinomial Logistic Regression* | | | | | | | | | | | | |
| | AME | p | AME | p | AME | p | AME | p | AME | p | | |
| *Discipline (Ref: LS)* | | | | | | | | | | | | |
| PE | .074 | .070 | .032 | .148 | -.082 | .006 | -.012 | .515 | -.011 | .652 | | |
| SH | .150 | .001 | .070 | .002 | -.131 | .000 | -.046 | .018 | -.042 | .079 | | |
| *PhD related characteristics* | | | | | | | | | | | | |
| Age at 1st job after PhD (centered) | .019 | .000 | -.019 | .000 | .001 | .988 | -.003 | .145 | .005 | .045 | | |
| PhD in home country | -.037 | .381 | .018 | .610 | .030 | .357 | .003 | .882 | -.014 | .579 | | |
| Leiden score of PhD Institution | .005 | .359 | .001 | .858 | .001 | .831 | -.005 | .088 | -.001 | .735 | | |
| *Age of youngest child when started 1st job after PhD (Ref.: no children)* | | | | | | | | | | | | |
| Under 3 | -.046 | .326 | .031 | .351 | -.011 | .743 | -.010 | .660 | .034 | .245 | | |
| 3 or older | .053 | .374 | -.039 | .388 | -.038 | .289 | .001 | .984 | .024 | .466 | | |
| Work exp. before PhD | -.024 | .499 | .019 | .476 | -.019 | .433 | .013 | .440 | .012 | .554 | | |
| *Personal characteristics* | | | | | | | | | | | | |
| Female (ref.: male) | .001 | .998 | -.075 | .018 | .040 | .203 | .023 | .326 | .011 | .655 | | |
| Born after 1960 | .315 | .000 | -.329 | .000 | .025 | .271 | -.003 | .822 | -.009 | .672 | | |
| *Nationality (ref: Central Europe)* | | | | | | | | | | | | |
| East Europe | -.071 | .341 | -.083 | .064 | .162 | .026 | .009 | .785 | -.017 | .556 | | |
| Southern Europe | -.070 | .123 | -.049 | .137 | .061 | .065 | .013 | .556 | .045 | .109 | | |
| Northern Europe | .050 | .421 | .049 | .350 | -.069 | .004 | -.028 | .230 | -.003 | .927 | | |
| Rest world | -.022 | .722 | .105 | .034 | -.049 | .115 | -.018 | .518 | -.016 | .602 | | |
| Pseudo R2 | .180 | | | | | | | | | | | |

Ref.: reference category, LS.: Life sciences, PS.: Physical sciences and Engineering, SH.: Social sciences and Humanities; control variables for no information for age at PhD and Leiden score of PhD institution; Anova test statistics †, Chi2 test statistics ‡ Sig.:

*** p<0.001.

** p<0.01.

* p<0.05. + p< = 0.1

are significant differences in career length. Furthermore, women account for about 15% of the sample and are significantly underrepresented in the *mature progress at universities* pattern. The grant success ranges from 9 per cent in the *steady progress in government* pattern to 16 per cent in the *mature progress at universities* pattern. Again, the differences are not statistically significant.

## Who follows which pattern?

We examined what factors influence whether a researcher follows one career pattern or another. For each pattern identified, we estimated average marginal effects (AME) from multinomial logistic regressions (Table 2 for StG and Table 3 for AdG). For categorical variables, the AME indicated by how many percentage points the probability of being in a certain pattern is on average higher or lower for a researcher with certain personal and PhD characteristics compared to the reference category. For example: In the StG sample (Table 2), female researchers have an 8 percentage point higher probability of being in the *steady progress in research institutes* pattern and a 10 percentage point lower probability of being in *quick advances in universities* than male researchers. For continuous variables, such as age and Leiden score, the AME indicated by how many percentage points the probability of being in a certain cluster increases (decreases) if the variable increases (decreases) by one unit. For example: In the StG sample, being older by one year when receiving the PhD is associated with a 2 percentage point increase in the probability of being in the *complicated moves across institutions* pattern.

**Starting grant applicants.** Given the role of institutions in deriving career paths, it is unsurprising that there are some discipline-based differences: researchers from the Life Sciences (LS) have a significantly higher probability of making *steady progress in research institutes* when compared to researchers from Social Sciences and Humanities (SH).

Those who were older when receiving their PhD are more likely to make *complicated moves across institutions*, and less likely to be in the *quick advances at universities* pattern. Moreover, researchers who had other work experience before commencing their PhD are less likely to make *quick advances at universities*. These findings indicate that there are path dependencies between fast progression towards the PhD and quick advances in the career after PhD. Presence in the *complicated moves across institutions* pattern is negatively correlated with the prestige of the institution from which researchers received their PhD. Whether researchers have already been internationally mobile during their PhD or not does not make a significant difference in terms of career pattern. Finally, compared with those who were not parents at the time of completing their PhD, those with older children (over three years) at the time of receiving the PhD have a lower probability of being in *steady progress at universities* and a higher likelihood, only significant at the 10 percent level to be in the *delayed advances at universities* pattern.

Regarding personal characteristics, scientists from the later birth cohorts, born after 1970, are less likely to be making *quick advances in universities*, and more likely to be in the *delayed advances in universities*, the *steady progress in research institutes*, as well as in the *complicated moves* pattern. Women are less likely than men to make *quick advances in universities*. Furthermore, women are more likely to be in the *steady progress in research institutes* pattern than men (only significant at a 10 percent level). This difference is not only linked to the high proportion of Life Sciences (LS) in research institutes, where women are proportionally overrepresented, but also to a generally higher likelihood of women to be employed in research institutes than men.

We also examine whether the careers of male and female scientists tend to develop differently by estimating the multinomial logistic regressions with interaction terms between gender

and PhD-related characteristics (results presented in S3 File). The results indicate whether PhD-related characteristics make differences in the probability of being in a certain pattern for a male and female researcher. There are few gender differences. Research prestige of the PhD institution based on the Leiden score is a significant factor for men only, in particular for their likelihood to be making *complicated moves across institutions*. Those who were parents at by the time that they completed their PhD are generally less likely to be in the *steady progress at universities* pattern. However, having care responsibilities for children over the age of three when receiving the PhD is a stronger factor for women. It is associated with a lower likelihood of women of making *complicated moves* cluster as well as increasing the likelihood of making *delayed advances at universities* compared to the women in the sample who are not mothers.

## Advanced grant applicants

There is a significant difference between patterns regarding research discipline. Given the predominance of LS research undertaken in research institutes, it is unsurprising that we again observe a higher likelihood of life scientists making *steady progress at research institutes*. SH is more dominant at universities so that social scientists or humanities scholars have a higher probability than those in LS of being in the *steady* or *mature progress at universities* pattern and a lower probability of being in all other patterns.

Those who were older when starting their first job after their PhD are more likely to make *complicated moves across institutions*, as reported for the StG sample. Furthermore, there are differences in the likelihood of being in *mature* or *steady progress at universities* observed by age of completing PhD, given otherwise equal age as controlling for birth cohort. Those in the *mature progress in universities* pattern in general were younger when starting their scientific career in contrast to those making *steady progress in universities*. Birth cohort, not surprisingly, is an additional differentiating factor between those having made *mature progress* at universities and those who have not. Moreover, *steady progress in government* is negatively correlated with the prestige of the institution from which researchers received their PhD (significant at 10 percent level). Neither international mobility during the PhD, work experience before receiving the PhD nor having children at the start of the career after PhD are significant factors in the likelihood of being in a particular career pattern. There is some significance for parenthood at the time of PhD. Finally, we find some gender differences as women, *ceteris paribus*, are less likely to be making *matured progress at universities*.

There are hardly any gender-specific relationships between relative early parenthood and career patterns (results presented in S3 File). Women researchers who had older children when starting their first job after the PhD are less likely to be making *steady progress in government* when compared to researchers who are not parents.

## Discussion and conclusion

Using sequence analysis of self-reported career histories of ERC applicants, we have identified multiple and distinct career patterns that represent combinations of positional and institutional sequences, different progression logics, and movements. Our contribution responds to the gap in the empirical literature, and the need expressed by policy makers and the broader scientific community [1], by mapping research careers and providing evidence-based insight into not only the variety in research careers but also into the breadth of institutional environments in which research is undertaken–thereby challenging conventional wisdom on research careers in the European context. Our results confirm that cumulative upward mobility is (still) the norm for research careers. In our study this is reflected in the predominant *steady progress* career patterns. However, the 'road to excellence' cannot be characterized only by this

traditional pattern–as conventions would have it. We found divergent career patterns including *complicated moves* that do not follow conventions of smooth progress. In particular among early career researchers in the Starting Grant sample (StG), differences in career patterns reflect differences in timing as illustrated by *quick* versus *delayed advances*. This variety in research careers is visible in our sample of applicants to the most prestigious individual research grant scheme in Europe. While the proportion of funded versus non-funded applicants is not the same across patterns, grantees are found in each; therefore one of our key results is that excellence in terms of ERC grant success is found across all career patterns. Both the variety of patterns and the presence of grantees across all patterns add to the validity of our findings. Even if based on a narrow population (because only an elite group of potentially excellent researchers applies for this kind of competitive funding), we have a broad sample that is representative of the ERC applicant population covering applicants from all disciplines, from EU and non-EU countries, and including both early and later career stages. A different sample may be distributed differently across patterns but would only produce limited additional patterns.

Across both samples of early and established researchers, we have identified two conventional and common career patterns of *steady progress* in universities or research institutes and a third, less conventional pattern, of *complicated moves* across institutions. In addition we find three career patterns that are uniquely related to the career stage: *quick* and *delayed advances* for researchers applying to the Starting Grant (StG) and *mature progress* for the Advanced Grant (AdG)–all within universities. The pattern of *steady progress in government* appears only for the AdG, but forms part of the *complicated moves across institutions* pattern for the StG. *Steady progress* is thus more common for AdG than StG, reflecting not only the more exclusive nature of this sample of established researchers but also career length–enough time has passed to detect steady progress. Although we observe cohort effects, *delayed progress* in universities or *steady progress* in research institutes are more common than the 5 other patterns amongst those born after 1970 and *steady progress* in universities is more common that *mature progress*. Our robustness analysis of the pooled samples (see S3 File) suggests that the differences in the appearance and the frequency of patterns between the StG and the AdG are to a large extent an age or tenure effect, meaning that those in the StG patterns will develop towards the equivalent AdG patterns over time.

Positions (and moves between positions) are more important in differentiating between patterns than institutions. Our parallel analysis of job positions and institutional affiliation shows that career progression primarily means changing positions, while movements across institutions are less common. One exception is the pattern of *complicated moves* across both positions and institutions. Institutions thus host research careers, and where careers develop (inside which kind of institution) is often a matter of discipline. We also see that different institutions host similar career patterns of *steady progress*–universities, research institutes and government—a finding that extends our understanding of research careers beyond those in universities.

In contrast to the prominent assumption that women's careers in research develop differently from men's, gender in itself makes little difference in terms of which career patterns men and women follow. Women among the StG applicants are less likely to be in the quick advances cluster; the small numbers of women among the AdG applicants are less likely to have achieved mature progress in universities. There are some indications that having children at the time of PhD affects men and women's career differently and that differences are more pronounced in the StG sample. One possible explanation for this finding is that the AdG sample is more selective as a consequence of low(er) representation of women in the older cohorts. The intersection of career mobility, children's ages, and timing of funding [63] is something

that deserves further exploration. With more detailed information on family formation and partners' careers available, sequence analysis could be applied to a joint analysis of work and family trajectories (e.g.; [56]), to explore the interlocked nature of family patterns and research career patterns. The fact that we find only very limited gender differences in career patterns, undermines the common assumption held by policy makers and contradicts the (limited) empirical evidence that careers of men and women in research develop differently. However, this could be an effect of the exclusive nature of our sample of ERC applicants. If career patterns do not differ between men and women applicants, but success rates in research funding do, we must reconsider the importance of CVs and gendered assumptions in selection decisions.

Our analysis also shows that discipline matters for career patterns. When looking at the researchers in each pattern, it is clear that those following careers in research institutes are typically from the Life Sciences. Path dependence makes a difference in terms of following particular career patterns. Prior work experience, age when receiving their PhD, PhD obtained from different prestigious institutions, and having children at the time of PhD are differentiating factors in the StG sample. However, we were not able to explore disciplinary differences within patterns, nor could we take underlying social and economic factors related to host country, country of origin, or international mobility (other than moving to do the PhD) into account. Evaluators looking for "excellence only" use career signals from applicants' CVs including mobility as proxies–a pattern of moves across institutions may be viewed positively when it includes various prestigious institutions across national borders. A route for future research would be to examine career patterns as an individual predictor of grant application success, alongside other personal and prestige indicators. Another would be to examine the stability of patterns both within and between patterns by extending the analysis to further cohorts of grant applicants and by following the StG applicants over time to see whether they continue the same trajectory in the future. The further funding and careers of those who applied but were rejected could also be examined. This would also shed light on the complex interactions of grant funding on the national and European level [21], as well as career consequences of reaching the quality threshold but not getting funded [24,29]. From a policy perspective, it would also be interesting to study the level of institutional support and the degree to which institutions discourage or even deny researchers the opportunity to apply for an ERC grant, something that may have affected the selectivity of our sample. In Spain, for example, universities' commitment to ERC "values of excellence" varies from evident to neglected [21].

Methodologically, our study has limitations but also opens possibilities. Using a survey to capture full career histories may affect response rates and thus limit coverage of an already selective sample in terms of career patterns identified. Sophisticated methods to extract information from CVs as submitted alongside applications have since been developed and tested, which could be used for further research [64]. A more structured CV format used in the application materials would certainly help in terms of consistency and comparability of career data. Sample size affecting statistical power, sample selection bias, and the analysis of only a single applicant cohort for the StG suggests caution in terms of generalizing our career pattern findings. However, the multichannel sequence analysis method we have used [59], could be used to identify career patterns among other samples of researchers or scientists, as well as other professions in which a common career start (e.g. initial professional qualification) and/ or ceiling (e.g. making partner) can be established.

This is the first application of sequence analysis to map contemporary European research careers across disciplinary, institutional, and national borders. We have shed light on career patterns in research and we provide a firm basis to explore implications of (un) conventional career patterns for grant application success of men and women in research. We hope our

findings on the occurrences and nuances of career patterns in research will inform policymaking, career development, mobility, and gender equality in the European Research Area.

## Supporting information

**S1 File.**
(DOCX)

**S2 File. Survey questions.**
(PDF)

**S3 File.**
(DOCX)

## Acknowledgments

We thank the School of Business & Economics at Vrije Universiteit Amsterdam for hosting the ERCAREER project. We appreciate the insightful and supportive comments from our reviewers, and we thank Dr. Christian Brzinsky-Fay (WZB) for his expertise in developing the patterns visualization,

## Author Contributions

**Conceptualization:** Claartje J. Vinkenburg, Sara Connolly, Stefan Fuchs, Channah Herschberg, Brigitte Schels.

**Data curation:** Claartje J. Vinkenburg, Sara Connolly, Stefan Fuchs, Channah Herschberg, Brigitte Schels.

**Formal analysis:** Claartje J. Vinkenburg, Sara Connolly, Stefan Fuchs, Channah Herschberg, Brigitte Schels.

**Funding acquisition:** Claartje J. Vinkenburg, Sara Connolly, Stefan Fuchs.

**Investigation:** Claartje J. Vinkenburg, Sara Connolly, Stefan Fuchs, Channah Herschberg, Brigitte Schels.

**Methodology:** Claartje J. Vinkenburg, Sara Connolly, Stefan Fuchs, Channah Herschberg, Brigitte Schels.

**Project administration:** Claartje J. Vinkenburg, Sara Connolly, Stefan Fuchs, Channah Herschberg.

**Resources:** Claartje J. Vinkenburg, Sara Connolly, Stefan Fuchs, Channah Herschberg, Brigitte Schels.

**Software:** Sara Connolly, Stefan Fuchs, Channah Herschberg, Brigitte Schels.

**Supervision:** Claartje J. Vinkenburg, Sara Connolly, Stefan Fuchs, Brigitte Schels.

**Validation:** Claartje J. Vinkenburg, Sara Connolly, Stefan Fuchs, Channah Herschberg, Brigitte Schels.

**Visualization:** Claartje J. Vinkenburg, Sara Connolly, Stefan Fuchs, Channah Herschberg, Brigitte Schels.

**Writing – original draft:** Claartje J. Vinkenburg, Sara Connolly, Stefan Fuchs, Channah Herschberg, Brigitte Schels.

**Writing – review & editing:** Claartje J. Vinkenburg, Sara Connolly, Stefan Fuchs, Channah Herschberg, Brigitte Schels.

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
