## [Decision Letter · Decision Letter 0]

17 Jan 2020

PONE-D-19-29861

Mapping Career Patterns in Research: A Sequence Analysis of Career Histories of ERC Applicants

PLOS ONE

Dear Dr Vinkenburg,

Thank you for submitting your manuscript to PLOS ONE. After careful consideration, we feel that it has merit but does not fully meet PLOS ONE’s publication criteria as it currently stands. Therefore, we invite you to submit a revised version of the manuscript that addresses the points raised during the review process.

The reviewers were highly favorable, but identified a few minor corrections that could improve the manuscript. We ask that you attend to these in a minor revision.

We would appreciate receiving your revised manuscript by Mar 02 2020 11:59PM. To enhance the reproducibility of your results, we recommend that if applicable you deposit your laboratory protocols in protocols.io, where a protocol can be assigned its own identifier (DOI) such that it can be cited independently in the future. For instructions see: http://journals.plos.org/plosone/s/submission-guidelines#loc-laboratory-protocols

We look forward to receiving your revised manuscript.

Kind regards,

Cassidy Rose Sugimoto, Ph.D.

Academic Editor

PLOS ONE

Journal Requirements:

4. Please include your tables as part of your main manuscript and remove the individual files. Please note that supplementary tables (should remain/ be uploaded) as separate "supporting information" files

Additional Editor Comments (if provided):

Reviewers' comments:

Reviewer's Responses to Questions

**Comments to the Author**

1. Is the manuscript technically sound, and do the data support the conclusions?

Reviewer #1: Yes

Reviewer #2: Yes

2. Has the statistical analysis been performed appropriately and rigorously? 

Reviewer #1: Yes

Reviewer #2: Yes

3. Have the authors made all data underlying the findings in their manuscript fully available?

Reviewer #1: No

Reviewer #2: No

4. Is the manuscript presented in an intelligible fashion and written in standard English?

Reviewer #1: Yes

Reviewer #2: Yes

5. Review Comments to the Author

Reviewer #1: This is a review of Mapping Career Patterns in Research: A Sequence Analysis of Career Histories of ERC Applicants.

This paper is straightforward, clear, reproducible (provided that the authors' offer to provide survey data to those interested), and interesting. The authors take careers trajectories, as reported by ERC applicants on a survey, and map them to sequences. The sequences are then compared pairwise and clustered into canonical career trajectories, making it then straightforward to ask questions like: are there differences in the demographic makeup of sequence clusters, or, do applicants from a particular cluster tend to get funded more often than others.

Below, I have some brief suggestions for clarity, but these are merely suggestions. The paper, as it stands, is recommended for publication. I'd like to thank the authors for writing something so clearly without attempting to overstate or understate their conclusions. It was a pleasure to read and review.

My only question is whether the survey data's limited availability complies with the PLOS data access policy. I leave it to the editor and authors to decide that.

// Suggestions:

1. After saying "process outcomes" in quotation marks, it would be good to explain to the reader what this means.

2. Why are there {reference withheld} indications, and will they be included in the published version?

3. "Since 2013, the Starting and Consolidator are separate grant schemes but at the time we collected our data, this was still one call." I'm not sure what "this was still one call" means. Can this be stated more clearly?

4. "this project was commissioned by the ERC gender balance thematic working group to explore gender aspects in career structures and career paths of applicants." I wouldn't mind knowing this purpose earlier. It places the access to data and reason for the study in context.

5. Can the figures be made larger or rearranged somehow? Even zooming in, it was hard to read them, but they are important to the findings of the paper, and are presented/formatted nicely. Given that there are not other figures in the paper, I'd suggest taking up a bit more space!

6. For figure captions, the non-abbreviated StG and AdG might be nice so that the skimming reader, problematic though such a reader may be, can dip in and still understand these punchline figures.

7. The paragraph on pages 14-15 that describes the demographic differences between the career patterns by success rate and gender lead with differences, but then state that those differences are not significant. This statement of "there are differences but they are not actually differences" is a little confusing, and I would suggest, if the authors feel comfortable, changing wording so that it is clear that it "appears at first glance" that there are differences or something like that, so that the language better reflects the statistical conclusions.

8. "linked to the high proportion of LS in research institutes"

By the time I got to this point, I had sort of lost the thread on the Life Sciences abbreviation. No need to make the decision based solely on my comments, but do the acronyms buy that much space saving or clarity?

9. Overall I thought that the references were appropriate and thorough. Nothing to add.

// Trivial changes or typos:

Therefore, our potential sample comprised of applicants

->

Therefore, our potential sample comprised applicants

second, the substation costs for

->

second, the substitution costs for

Reviewer #2: The paper "Mapping Career Patterns in Research: A sequence Analysis of Career Histories of ERC Applicants" by Vinkenburg, Connolly, Fuchs, Herschberg and Schels presents a survey-based analysis, categorizing career patterns by their progression through career stages (sequences) and locations. They follow up the categorization with an analysis of the parameters describing who follows which pattern.

The study is innovative in the way data and analysis is combined, and it is a highly relevant topic. I applaud the authors for a well-planned research design.

I would be happy to recommend this paper for acceptance, but have a few minor comments, which the authors might consider as possible improvements to their paper. I also need to know why some of the references are withheld - it is not clear from the context why this is the case, or if they will be included at the time of publication.

Suggested revisions:

I appreciate the review section. However, it is at times a little bit superfluous in that it summarizes a number of studies very briefly without explicitly using that information for anything. It almost gives this reader the impression that those references are included more "for show". This is not something I want to think of an otherwise great paper, so I would suggest that the authors carefully consider if a) all references in the review are important to include and b) if they could add something more conclusive or argumentative to the more summarizing parts of the review.

I also suggest expanding a little on the methods, especially OMA. It is good that the authors include references to previous uses and tests of OMA, but it would be great to see a slightly more technical explanation of the method, e.g. in terms of how observations are coded and compared, or alternatively an example of the matching using made-up data, that allows the reader to more clearly understand the procedure.

Ward clustering is also described very briefly, but as this is a somewhat more well-known approach, I find that more acceptable. But the authors could consider expanding this section slightly as well, perhaps using a visual explanation.

6. PLOS authors have the option to publish the peer review history of their article (what does this mean?). If published, this will include your full peer review and any attached files.

Reviewer #1: No

Reviewer #2: No

---

## [Author Response · Author response to Decision Letter 0]

23 Apr 2020

Thank you to our reviewers - our response is addressed in the rebuttal letter

---

## [Decision Letter · Decision Letter 1]

6 Jul 2020

Mapping Career Patterns in Research: A Sequence Analysis of Career Histories of ERC Applicants

PONE-D-19-29861R1

Dear Dr. Vinkenburg,

We’re pleased to inform you that your manuscript has been judged scientifically suitable for publication and will be formally accepted for publication once it meets all outstanding technical requirements.

Kind regards,

Ting Ren

Academic Editor

PLOS ONE

Additional Editor Comments (optional):

The authors have addressed sufficiently the reviewers' concerns and suggestions; therefore, I am pleased to accept the paper.

Reviewers' comments:

Reviewer's Responses to Questions

**Comments to the Author**

1. If the authors have adequately addressed your comments raised in a previous round of review and you feel that this manuscript is now acceptable for publication, you may indicate that here to bypass the “Comments to the Author” section, enter your conflict of interest statement in the “Confidential to Editor” section, and submit your "Accept" recommendation.

Reviewer #2: All comments have been addressed

2. Is the manuscript technically sound, and do the data support the conclusions?

Reviewer #2: Yes

3. Has the statistical analysis been performed appropriately and rigorously? 

Reviewer #2: Yes

4. Have the authors made all data underlying the findings in their manuscript fully available?

Reviewer #2: No

5. Is the manuscript presented in an intelligible fashion and written in standard English?

Reviewer #2: Yes

6. Review Comments to the Author

Reviewer #2: (No Response)

7. PLOS authors have the option to publish the peer review history of their article (what does this mean?). If published, this will include your full peer review and any attached files.

Reviewer #2: **Yes: **Jens Peter Andersen

---

## [Editor Report · Acceptance letter]

15 Jul 2020

PONE-D-19-29861R1 

Mapping Career Patterns in Research: A Sequence Analysis of Career Histories of ERC Applicants 

Dear Dr. Vinkenburg:

I'm pleased to inform you that your manuscript has been deemed suitable for publication in PLOS ONE. Congratulations! Your manuscript is now with our production department. 

Kind regards, 

on behalf of

Dr. Ting Ren 

Academic Editor

PLOS ONE